# Review of Miniaturized Computational Spectrometers

**DOI:** 10.3390/s23218768

**Published:** 2023-10-27

**Authors:** Qingze Guan, Zi Heng Lim, Haoyang Sun, Jeremy Xuan Yu Chew, Guangya Zhou

**Affiliations:** Department of Mechanical Engineering, National University of Singapore, Singapore 117575, Singapore; qingze.guan@u.nus.edu (Q.G.); limziheng@nus.edu.sg (Z.H.L.); sunhaoyang@u.nus.edu (H.S.); j.chew@nus.edu.sg (J.X.Y.C.)

**Keywords:** computational spectrometer, nanophononics, compressive sensing

## Abstract

Spectrometers are key instruments in diverse fields, notably in medical and biosensing applications. Recent advancements in nanophotonics and computational techniques have contributed to new spectrometer designs characterized by miniaturization and enhanced performance. This paper presents a comprehensive review of miniaturized computational spectrometers (MCS). We examine major MCS designs based on waveguides, random structures, nanowires, photonic crystals, and more. Additionally, we delve into computational methodologies that facilitate their operation, including compressive sensing and deep learning. We also compare various structural models and highlight their unique features. This review also emphasizes the growing applications of MCS in biosensing and consumer electronics and provides a thoughtful perspective on their future potential. Lastly, we discuss potential avenues for future research and applications.

## 1. Introduction

Spectrometers are critical instruments in analytical applications and have demonstrated extensive usage across diverse fields, such as chemistry [1], biology [2], food science [3], and agriculture [4]. The evolution of spectrometry began with rudimentary dispersers such as prisms, which date back to the era of Newton [5], and has since progressed to a more compact and efficient design. Contemporary designs typically incorporate fore-optics and a grating [4], allowing for wavelength-sensitive light dispersion after transmission through the grating. Subsequently, individual wavelengths can be registered and analyzed.

Classically, disperser-based spectrometers function by demultiplexing the incoming light into its constituent wavelengths. This enables direct measurement of the intensity of each wavelength, thus forming the spectrum of the input light. However, the resolution of disperser-based spectrometers is usually constrained by aberrations within the optical system, the pixel size of the camera sensor, and manufacturing discrepancies. Moreover, the throughput of the system is limited due to the necessity of eliminating spatial information, leading to a compromised signal-to-noise ratio (SNR).

The recovery of spectra through computational methods seeks to simplify the spectrum acquisition system while improving both spectral resolution and SNR. Notable examples include the Fourier transform spectrometer (FTS) [6] and the Hadamard transform spectrometer (HTS) [7,8]. Many FTS operate on the principle of the Mach–Zehnder interferometer (MZI), adjusting the interference of the input light to capture the Fourier transform of the spectrum. There are several mature FTS commercially available, with further discussion of FTS to follow later in this review. Conversely, HTS employs multiple encodings to capture the spectrum using a single-pixel detector, offering a potential cost reduction. Nevertheless, the alignment issue remains a significant challenge in FTS, and research is ongoing to address it.

While spectrometers hold great utility, their conventional setups are often bulky and expensive [8], limiting their applications in certain scenarios. Development in nanophotonics manufacturing technology and the advent of intensive light-matter interactions in silicon devices have enabled the application of spectrometry to chip-wise systems. This significant advancement has drastically reduced device size, broadening the scope of applications for miniaturized spectrometers.

It is possible to develop micro-spectrometers based on miniaturized diffractive optics. However, such a design remains subject to the issues faced by disperser-based spectrometers, including aberrations and alignment. Therefore, it is preferable to utilize nanophotonic structures as light modulators. Directly recording the intensity of each wavelength could require a large device footprint; hence, measurement points are typically fewer than the data points obtained with the aid of compressive sensing (CS) theory. The CS theory states that a signal can be largely recovered with fewer samples than stipulated by the Nyquist sampling theory with minimal loss. Under CS theory, the spectrum is encoded with a nanophotonic structure, then decoded using computational methods. The integration of nanophotonics technology and computational methods gives rise to miniaturized computational spectrometers (MCS).

In recent decades, significant research effort has been devoted to MCS. Numerous effective structures and encoding schemes have been proposed. To support ongoing and future research, this review summarizes prior work, comparing them to outline the present research landscape and gaps. Since we focus on delivering a comprehensive understanding of this field, not much space is dedicated to non-computational spectrometers and hyperspectral imaging systems. A number of valuable research studies exist on miniaturized spectrometers based on traditional methods as well as computational hyperspectral imaging systems [9,10,11,12], which expand the analysis from 1D spectral information to 3D data cubes. However, this paper focuses on MCS only, and we direct interested readers to other informative reviews, such as [13,14,15,16,17].

The review is structured as follows: Section 2 outlines the basic principles of all the MCS discussed herein, describing the common encoder–decoder structures. This serves as the fundamental theory of MCS. Section 3 compares various encoding strategies and discusses their advantages and disadvantages. Representative works are highlighted and analyzed. Section 4 explores the decoding algorithms for MCS, including convex optimization and machine learning. Several recent algorithms are presented. Section 5 offers examples of the wide range of applications of MCS, and Section 6 concludes this review by discussing the future direction and outlook of MCS.

## 2. General Principles of Miniaturized Computational Spectrometers

### 2.1. Architecture of Miniaturized Computational Spectrometers

The main feature that sets computational spectrometers apart, in addition to their small size achieved through nanophotonics, is the special kind of signal they record. In contrast to traditional spectrometers that measure the intensity of each wavelength directly, computational spectrometers capture an encoded version of the spectrum. Modulation is applied to the spectrum, and the captured signal represents a composite result of the modulation. It is then decoded computationally.

Hence, the essence of a computational spectrometer is constituted by two primary components: the encoder and decoder. The simplest encoder is represented by an array of narrowband filters, each of which transmits only a single wavelength, thereby recording all wavelengths sequentially. This can be expressed as follows:(1)$$y_{i}=\underset{\lambda }{\sum }a_{i}\left(\lambda \right)x\left(\lambda \right),i=1,2,3,\ldots ,n$$

Here, λ is wavelength, $x\left(\lambda \right)$ is the spectrum to be measured, $a_{i}\left(\lambda \right)$ is the transmission function for the *i*^th^ measurement, and $y_{i}$ is the *i*^th^ measured signal. This can be written in a matrix format:(2)Y=AX
where Y is the matrix containing the measured signals, A is the matrix containing the transmission function $a_{i}\left(\lambda \right)$ in each row, and X is the actual spectrum. This format applies to all the MCS scoped in this review. The narrowband filters [18] can be conceptualized as an encoder forming a diagonal matrix, with all diagonal elements being 1:(3)$$A_{n\times n}=\left[\begin{array}{ccc} \begin{array}{ccc} 1 & 0 & 0\\ 0 & 1 & 0\\ 0 & 0 & 1 \end{array} & \cdots & 0\\ \vdots & \ddots & \vdots \\ 0 & \cdots & \begin{array}{ccc} 1 & 0 & 0\\ 0 & 1 & 0\\ 0 & 0 & 1 \end{array} \end{array}\right]$$

Here, $\;A_{n\times n}$ is the transmission matrix where n is the number of measurements and the number of datapoints recovered. However, such an approach necessitates a large number of filters, which considerably increases the footprint of the device. Also, its SNR is limited due to the majority of the energy not being received with the detector since only one wavelength is transmitted each time. Furthermore, as resolution increases, so does the manufacturing precision required to maintain the central peak of each filter. Consequently, the device becomes more fragile, requiring stability in the face of environmental changes, limiting the utilization of narrowband filters.

In contrast, broadband filters used in computational spectrometers allow all the filters to respond to the entire input band [19,20,21]. Thus, the output of each filter represents a summation of the product of the response curve and the input spectrum, effectively sampling the spectrum by varying weight combinations for each wavelength. The recorded signal corresponds to the receptor number n, which should typically be significantly smaller than the number of data points in the input spectrum m. The transmission is now $\;A_{n\times m}$ as shown below, with $a_{nm}$ referring to each element in the matrix, most of which are non-zero. The problem is underdetermined and cannot be directly solved, so CS is needed.
(4)$$A_{n\times m}=\left[\begin{array}{ccc} \begin{array}{ccc} a_{11} & a_{12} & a_{13}\\ a_{21} & a_{22} & a_{23}\\ a_{31} & a_{32} & a_{33} \end{array} & \cdots & a_{1m}\\ \vdots & \ddots & \vdots \\ a_{n1} & \cdots & a_{nm} \end{array}\right]$$

This section breaks computational spectrometers down into two components: the encoder and decoder. Despite their common theoretical basis and common methods of convex optimization for signal recovery, the encoding process remains crucial in device design. Leveraging intricate light-matter interactions, we have access to various encoding methods. Thus, this review focuses on presenting a comprehensive examination and comparison of the primary encoding methods.

### 2.2. Compressive Sensing

The development of MCS is closely linked with CS theory. The CS theory posits that if a signal exhibits sparsity in a certain domain, fewer samples are needed to retrieve most of the signal’s information through optimization methods. The sampling ratio is typically significantly lower than that prescribed by the Nyquist sampling theorem, leading to an underdetermined problem [22]. The CS theory requires fewer measurements than the data points to be recovered, and as such, the footprint of MCS is significantly reduced. In addition to sparsity, CS theory necessitates that the sensing matrix be incoherent [23]. And with all requirements fulfilled, a large portion of the signal can be recovered via convex optimization [24].

Prior to the advent of CS-based methods, which employ regularization with optimization, researchers had explored other strategies for solving underdetermined problems, including singular value decomposition (SVD) and dictionary learning. These will be further expounded in Section 4.

## 3. Encoder

Before explaining the different designs of encoders, it is essential to understand the governing principles that influence the design of various encoding methods. The main factors that need to be considered in this context include footprint, resolution, and bandwidth. Typically, MCS balance between these three parameters. This is primarily due to the constraint that the recovered data points are limited, thereby creating a situation requiring a trade-off. If the focus is directed toward a specific region, the resolution is increased at the expense of bandwidth, and vice versa. Similarly, the footprint imposes restrictions on the data points collected, adding another aspect to this balance.

### 3.1. Fiber

A direct design of modulators for computational spectrometry is the utilization of multimode fibers (MMFs) as encoders. Hang et al. [25] pioneered this approach by integrating a bundle of photonic bandgap fiber alongside a monochrome charge-coupled device (CCD) to make a spectrometer. The transmission curves of the fibers differ from each other. Their experiment yielded a resolution constraint at approximately 30 nm.

Subsequently, Redding et al. [26] ventured further by employing solely an MMF for spectrometric applications. Inside the MMF, wavelength-dependent speckles are manifested due to inherent interference between different wavelengths. As varied wavelengths are channeled into the fiber, the emergent pattern is an encoded version of these distinct wavelengths. By using optimization techniques, spectrum reconstruction becomes feasible. Also, the spectral response sharpens with an increase in the length of the fiber. A notable experiment with a 20 m long fiber culminated in an impressive resolution of 8 pm. Furthermore, the ability to measure a broadband light spectrum was retained with a fiber length of merely 2 cm, albeit with diminished resolution.

Later innovations [27] led to the use of an ultra-long 100 m fiber, remarkably miniaturizing the scale of the device. The device is shown below in Figure 1a. With this configuration, they achieved a high resolution of 1 pm at a wavelength of 1500 nm. This represents a large improvement in resolution, though with a significant trade-off in cost. Also, it is analyzed that variation in ambient temperature, a factor that may influence the performance of fibers, can be largely rectified through recalibration.

However, a persistent challenge with MMFs is the inherent trade-off between resolution and bandwidth. Although enlarging the fiber core permits the passage of a broader band of spectrum, this invariably dilutes the contrast between speckles across these bands, which diminishes the spectrometer resolution. Addressing this problem, a subsequent experiment employed a bundle of seven MMFs [28], complemented by commercial wavelength division multiplexers (WDM). This setup allowed simultaneous multiplexing of signals into each MMF and concurrent capture of outputs across all fibers, as shown in Figure 1b. The outcome was a substantial expansion of bandwidth to 100 nm, coupled with a commendable 0.03 nm resolution at a wavelength of 1500 nm. Demonstrating the real-world applicability of this advancement, they illustrated its potential in spectral optical coherent tomography (OCT).

Fiber spectrometers offer multiple advantages, including broad bandwidth, cost-effectiveness, and a simplified manufacturing process. Yet, even with a noteworthy reduction in size [27], they remain less compact compared to on-chip devices. Therefore, there is potential for further reductions in their footprint.

### 3.2. Waveguide

Waveguides have been extensively employed in nanophotonics [29]. Similar to fibers, a multimode waveguide can be used directly as a spectrometer, given that the modes within elicit a wavelength-dependent response arising from the interference between diverse wavelengths. It has long been used in spectrometers [30,31,32]. Nevertheless, it is necessary to elongate the optical path (OP) to achieve superior resolution, which consequently amplifies the footprint of the device. This invariably impedes miniaturization, rendering it a predominant limitation. To circumvent this challenge, several strategies have been proposed.

A straightforward approach entails a waveguide array [33]. Schmid et al. [34] presented MCS constructed from an array of 50 waveguides, characterized by diminutive waveguide apertures. This configuration constrained the dimensions of the device to 8 mm × 8 mm. Wijk et al. [35] integrated arrayed waveguides with a multi-input multimode interference coupler to achieve an expansive bandwidth. In a similar vein, Zhang et al. [36] introduced a scanning mechanism by using arrayed waveguides with a micro-ring resonator. This innovation promises large bandwidth and high resolution, rendering it apt for OCT applications. Successive research includes experiments with tandem arrayed waveguide and micro-ring resonator configurations [37]. In [38], a tandem structure of a ring resonator and arrayed waveguides is used, and the scheme is shown in Figure 2a. Additional mechanisms have also been broached in a subsequent study [39], where 40 arrayed waveguides are compactly combined to increase the resolution.

An alternative strategy lies in coiling the waveguide to pack an extensive OP within a compact footprint. Previously, this coil methodology has found applications in diverse fields, notably sensing [41], and others [28,29]. Redding et al. [42] proposed a spiral waveguide as a spectrum modulator. This innovative Archimedean coiling considerably augmented the length of the OP. In addition to the spiral configuration, they utilized evanescent coupling, which operates across a broad band, thereby enhancing resolution. The challenge of mode mixing was mitigated through the speckle pattern generated via evanescent coupling. Their innovation achieved a resolution of 10 pm at a wavelength of 1520 nm, all encapsulated within a device merely 250 µm in radius.

Stratified waveguides have been utilized as broadband filters to establish the transfer matrix [40]. Figure 2b shows the layout. Given that a waveguide is an adaptable 3D structure, its morphology can be tailored to exhibit varied responses to different wavelengths. Therefore, crafting waveguides with distinct shapes emerges as a direct methodology for constructing MCS. Li et al. [40] employed this architectural paradigm to create a stratified array of filters. The design rationale for the structure centered on minimizing the correlation of identical wavelengths across various filters. The chosen parameters were devised to be compatible with optical lithography. Although certain ripples manifested on the device, their impact remained relatively insignificant.

Further, the stratified structure can be repurposed as a switching network dedicated to wavelength selection. Subsequently, the network channels the wavelengths to the MMF to analyze each spectral range [43]. Venturing into the realm of diverse waveguide architectures, Hornig et al. [44] harnessed a tapered hollow waveguide, leveraging its dispersion to attain a high resolution of better than 10 pm. In a parallel endeavor, Civitci [45] conceptualized adiabatically connected slab waveguides, employing the slab to mimic a prism-like mechanism.

### 3.3. Random Structure

As mentioned above, a critical limiting factor for waveguide-based devices is the OP. In addition to innovative waveguide designs, random nanostructures offer an interesting alternative. By folding the OP, these nanostructures substantially increase the length of the OP. Consequently, such random configurations can serve as modulators for light, primarily since they expand the interior OP. In this context, spectral data are transmuted into a spatial speckle pattern, permitting the differentiation of wavelengths with their respective patterns. Calibration of the transfer matrix for such a random structure can provide a matrix of random numbers, capturing the spectral-to-spatial transformation.

A noteworthy feature of random structures is their propensity for multiple scattering, which elongates the OP across all wavelengths, thereby enhancing resolution [46]. As presented in [46], an on-chip random structure was designed in which light experienced random scattering upon entering the framework. Subsequent collection and direction to the detector were facilitated using a full-band gap photonic crystal (PhC). This random medium enhanced resolution, achieving 0.75 µm within a 1500 nm bandwidth while concurrently shrinking the device footprint to a mere 25 µm radius. However, a challenge arises due to the potential for significant losses attributed to out-of-plane scattering within these random structures. To overcome this challenge, the loss was carefully analyzed and reduced by calculating the structural correlations inherent to the medium. The implementation of an amorphous structure led to a boost in optical transmission, exceeding twice the original capacity.

Randomized structures offer a novel paradigm in light modulation. However, their utility has mostly been limited to the telecommunication spectrum due to the intrinsic transparency parameters of silicon [47]. Addressing this spectral limitation demands innovative methodologies. A promising solution was proposed by Hartmann et al. [47], who utilized a silicon nitride (Si_3_N_4_) substrate, celebrated for its broad transparency window. This apparatus showcased operational capabilities ranging from the visible to the telecommunication bands, with its resolution intricately linked to the mean free path [48].

However, a notable drawback of these nanostructures is their increased sensitivity to environmental conditions. To address this issue, Sun et al. [49] developed an advanced system that incorporates a nano-void design to specifically reduce environmental sensitivity in scattering phenomena. Furthering this innovation, Sun et al. [50] synergized random structures with traditional waveguides, thus eliminating the requisite for conventional power-splitting mechanisms, such as Y-junction trees.

Despite these advancements, there are still more challenges. A predominant issue is the inherent variability in manufacturing these random structures, which directly impinges on the resultant resolution. Recent research [51] has identified the existence of eigenchannels within these configurations. Notable for their improved transmission efficiency compared to alternative methods, these findings offer valuable insights for future nanostructure design efforts. Instead of a stochastic design ethos, a more calculated, optimization-driven approach may yield superior modulation architectures. A comprehensive examination of these inverse nanostructure design intricacies will be conducted in subsequent discussions.

Beyond the realm of artificial random designs, natural structures are also inspirational. An avant-garde illustration of this is proposed by Kwak et al. [52], who leveraged the intrinsic structure of natural pearls to make a spectrometer. With a structure that combines crystalline aragonite and organic macromolecules in a randomized, layered composition, pearls serve as effective light-modulating agents. This bio-inspired modulation approach, followed by traditional spectral reconstruction methodologies, accentuates the promising horizon of employing inherently randomized, yet scalable, natural nanostructures.

### 3.4. Nanowire

Semiconductor nanowires, extending beyond their conventional applications, have found utility as modulators, echoing the functionalities of waveguides [53]. The academic foray into nanowires has primarily been enriched by the discipline of nanophotonics [54]. Contrary to the multi-dimensional structure inherent to waveguides, nanowires predominantly present a unidimensional architecture. An intriguing avenue explored within this domain is material alloying, which facilitates the construction of multifarious filters. By judiciously alloying diverse materials in varying proportions, an expansive and adaptive scheme can be conceived. While alloyed thin films present significant challenges stemming from intricacies in the fabrication processes [55], nanowires, whose manufacture is comparatively facile, demonstrate greater promise [56].

In the study shown in [56], an array of nanowires is strategically positioned as a modulator, interfacing directly with a sensor. A notable feature of this ensemble is the compositional gradient present within each nanowire, as demonstrated via the CdS*_x_*Se_1−*x*_ formulation. Such compositional nuances bestow each nanowire with a distinct response function. Light originating from free space undergoes modulation upon interaction with nanowires. Upon data acquisition with the receiver, an adaptive regularization methodology is employed to reconstruct the spectrum. Achievements from this study indicate an impressive resolution, reaching 15 nm or 10 nm using 30 or 38 nanowires, respectively, centered around a wavelength of 570 nm. Furthermore, the apparatus showed its potential in hyperspectral imaging and in situ examinations. With an ultra-compact design, the technology permits the integration of multiple such devices into an array, thereby facilitating hyperspectral imaging with minimal spatial requirements. However, a challenge exists because the response functions of different nanowires are similar to each other [17]. Such similarities can inevitably circumscribe the attainable resolution.

The unique structure of nanowires makes them well-suited for easy combination with sensors, a benefit effectively used in the research cited in [57]. Instead of using various methods to mix different metals together, the scientists used the concept of structural coloration [58]. Previous research has shown that changes in the diameter of the nanowires can lead to shifts in the wavelengths they absorb [59]. Building on this knowledge, the study deployed 24 pixels, wherein each pixel encompasses a periodic array of nanowires, each differing in radius. Thus, each pixel exhibits a unique transmission function for the spectrum. This setup allows the sensor to directly collect color or light information, eliminating the need for additional receiving devices. The design is shown in Figure 3b.

Recent progress in this area has shown that a single nanowire can be effectively used as a tool to measure the spectrum of light. Zheng et al. [60] employed a compositionally graded CdS*_x_*Se_1−*x*_ nanowire, which functioned analogously to a monolithic spectrometer. This arrangement was then used to measure light emission at a single point. Additionally, superconducting nanowires have emerged as catalysts for photon-counting detectors, showing great promise for spectrometer development. A notable endeavor by Cheng et al. [61] entailed the crafting of MCS using superconducting nanowires. They used dispersive grating along with single-photon counters. This system works from the visible to the infrared. Subsequent advancements by Kong et al. [62] strategically avoided the optical modulator, instead modulating the quantum efficiency of superconducting nanowires. This configuration achieved a commendable resolution of better than 10 nm and the capability for time-of-flight measurements. Complementing this, Zheng et al. [63] combined metasurfaces with photon-counting detectors, establishing empirical feasibility.

### 3.5. Photonic Crystal

PhC stands as a pivotal component within nanophotonics and has been extensively utilized in sensing applications [64]. Characterized by its periodic structure, a PhC modulates photons akin to the manner in which a conventional crystal modulates electrons [64]. Available in varied dimensions—1D, 2D, and 3D—the properties of a PhC can be modified by adjusting its structure and periodicity. This affords researchers a degree of flexibility in designing devices tailored to specific requirements [64]. Through the strategic design of PhC configurations, the creation of MCS becomes feasible [65].

Utilizing a PhC to construct a spectrometer involves employing the PhC slab as a light modulator, with the subsequent output captured with a CCD. Pervez et al. [66] demonstrated a device comprising a 3 × 3 grid of PhC slabs, wherein each cell exhibited distinct patterns and periods, thereby inducing varying spectral responses. This configuration decomposes the input spectrum using nine distinct response curves. Notably, the device is both compact and cost-effective and can be adapted to various applications. However, its resolution remains a limitation, approximately 40 nm within the visual range [67]. To increase resolution, Gan et al. [67] employed a similar conceptual framework, leveraging a high-quality factor semiconductor planar PhC as the modulator. This was strategically positioned atop a low-index waveguide composed of glass or polymer. Utilizing a gallium phosphide planar PhC array and a polydimethylsiloxane waveguide, they significantly enhanced the resolution to 0.3 nm at 840 nm.

The devices mentioned highlighted the potential of PhC as a spectrometer, but they encounter challenges in efficiency given that photons distribute uniformly and only a minimal fraction reaches the camera [68]. Meng et al. [68] advocated for a linear waveguide to eliminate this concern. Such a waveguide, adept at directing photons toward matched ports, augments device efficiency. Utilizing 24 PhCs, an efficiency of 6.36% was achieved, with a resolution of 1 nm at 1527 nm within the 1520–1545 nm band.

Prevailing studies typically engage PhCs as bandpass filters, ensuring distinct peak wavelengths for each PhC. However, innovations in CS and advanced manufacturing have propelled design paradigms. Wang et al. [69] utilized 400 PhC slabs, each characterized by unique patterns and periodicities. Consequently, the spectral response of each slab varies, displaying relative randomness across wavelengths. As light traverses the slabs, the outputs are recorded, with resolution contingent on the intricacy of response functions at specific wavelength intervals. A resolution of 0.1 nm was achieved from the sharp spikes. The design is shown in Figure 4a. Subsequent endeavors [70] led to the creation of an entire PhC slab integrated with various lattices, attaining single-shot spectrum recovery. This slab, encompassing a 6 × 6 PhC cell matrix, communicates its signals to a CCD, enabling easy transformation into a hyperspectral imaging system. Enhancement prospects include leveraging silicon nitride or silicon carbide, potentially yielding more intricate responses. The scholars further established a compact, single-shot spectral sensor, harmonizing a similar PhC device with complementary metal-oxide semiconductor (CMOS) technology.

An alternative approach entails the use of PhC nanobeams, mirroring methods seen in arrayed and stratified waveguides. While nanobeam cavities are primarily 1D structures, they have robust light confinement capabilities, further reducing the device footprint [71,72,73]. Cheng et al. [74] integrated a nanobeam cavity array to computationally reconstruct the spectrum, as illustrated in Figure 4b. Each nanobeam operates similar to a bandpass filter. The device has a core length of 6 µm and a spacing of 3 µm, making it relatively small in size. Its design allows for easy scaling, and because there is a known relationship between its operational frequency and its structural details, it is possible to adapt the device for different wavelengths. Tests by the researchers showed that the device works effectively in the 1310 nm, 1580 nm, and 2400 nm ranges. Later, Zhang et al. [74] utilized a cascaded structure nanobeam to enhance resolution and bandwidth.
Figure 4PhC-based MCS. (**a**) PhC slab layout (top) with each unit having a different response curve (right) and image for the device (bottom) (figure reprinted from [70] with permission © 2019 Nature Publishing Group); (**b**) pipeline of using PhC nanobeam as a modulator to make MCS. The device (top), transmission function (bottom), and algorithm (right) are shown here (figure reprinted from [73] with permission © 2021 American Chemical Society).
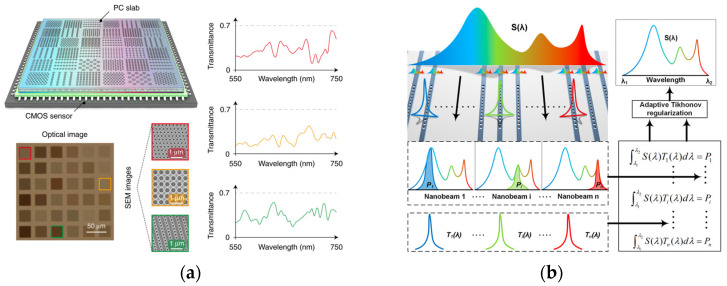



### 3.6. Quantum Dot

Quantum dots (QDs) are semiconductor nanocrystals characterized by their radii being smaller than the bulk exciton Bohr radius. Such dimensions result in the quantum confinement of electronic charges. Moreover, size variations in these QDs correspond to changes in bandgap [48,49]. Given their capacity to act as modulators and their compatibility with CCDs, QDs show potential for the development of miniaturized spectrometers. Bao et al. [75] pioneered this approach by employing QDs to construct spectrometers. They fabricated a filter using an array of 95 colloidal QDs, each varying in size and composition. The absorption properties of each QD were meticulously designed to evenly span the 390–690 nm wavelength range, with representative absorption profiles depicted below. The QD array facilitated the creation of numerous filters, bolstering resolution capabilities. The achieved resolution was 1 nm within their specified operational range, and the QDs demonstrated prolonged stability.

However, the resolution detailed in [75] remained constrained. Addressing this limitation, Zhu et al. [76] introduced perovskite QD films as alternative filters. With the inherent benefits of high transmission and repeatability, an array of 361 perovskite QD films was paired with a CCD to capture signals. This system encompassed a broad wavelength range, spanning 200–1000 nm.

Extensions into the near-infrared spectrum have also been achieved. Specifically, Li et al. [77] employed PbS and PbSe QDs to design a spectrometer operating within the 900–1700 nm wavelength range. Utilizing 195 QDs in conjunction with an InGaAs CMOS camera, they reported a resolution of 6 nm. While echoing the principles of [75], this design offered distinct advantages over prior near-infrared designs that employed narrowband filters and gratings. Analyses indicated that no significant resolution enhancements were observed beyond 180 QDs, leading to their choice of 195 QDs.

Several decoding methodologies designed for QDs have been introduced. For instance, Zhang et al. [78] advocated for a specific machine learning approach to more efficiently extract spectrum data from QD measurements. This integrated network, combined with solvers, outperformed traditional optimization and regularization techniques. In a subsequent development, an autoencoder was utilized to mitigate noise during the reconstruction phase [79]. Initial data denoising via the autoencoder were followed by reconstructions through sparse algorithms.

However, as highlighted in [75], spectral responses across individual QDs often exhibit similarities, which may complicate the generation of pronounced peaks. While enhancing resolution and bandwidth through additional QDs is feasible, it simultaneously escalates the footprint and cost of the device. Additionally, potential manufacturing errors and their implications remain unaddressed [75]. Nevertheless, QD-based spectrometers stand out due to their cost-effectiveness, adaptable spectral range, and compact design, positioning them as potential contenders in industrial applications. Initiatives to commercialize QD-based spectrometers are currently underway [80].

### 3.7. Metasurface

Extensive research on metasurfaces has been conducted, revealing their exceptional potential for modulating light with increased degrees of freedom and tunability [81,82,83]. The wavelength sensitivity of metasurfaces has been recognized, leading to their integration in MCS applications [84,85]. Furthermore, many metasurfaces employed in MCS are metal-based, prompting a convergence of these two categories.

In the study [85], a metasurface array functions as a light encoder. This sensor comprises an array of distinct metasurface units. The unique periodic structure of each metasurface unit dictates its respective response function. Changes in these structures allow for a variety of responses, making the device design more flexible. Light modulated with these metasurface units is subsequently converged using microlenses and directed to an image sensor, optimizing the signal capture on the sensor. The amalgamation of individual metasurface units into larger supercells culminates in ultraspectral imaging capabilities.

Additionally, these metasurfaces can be adjusted or reconfigured. In each larger unit, or super cell, the arrangement of the smaller cells can be regularly changed to improve the resolution for different settings. The resulting resolution is a notable 0.8 nm, and the small size of the ultra-detailed camera highlights its effectiveness. These improvements show that using metasurfaces in MCS designs is a practical choice.

Zhu et al. [86] harnessed metasurfaces to engineer dispersion, carefully controlling phase, group delay, and group delay dispersion. This approach allowed for the correction of aberrations in spectrometers, thereby eliminating the need for bulky optical components and streamlining the system. Similarly, as shown in Figure 5a, Faraji-Dana et al. [87] employed multiple metasurfaces to manipulate the OP and disperse light, resulting in a considerable reduction in the size of traditional spectrometers. Their innovative design achieved a resolution of approximately 1.2 nm within a size of merely 7 mm^3^. In another study, Hu et al. [88] integrated metasurfaces with thin films. Meanwhile, Zhang proposed a metalens design that enhanced beam focusing, collimation, and deflection for dispersive on-chip spectrometers [89]. The structure is demonstrated in Figure 5b. However, though metasurfaces have been extensively utilized in bulk-free space spectrometers, leading to improved resolution and compactness, they are still major free-space devices and have yet to achieve high miniaturization [90].

Certain metasurfaces capitalize on the plasmonic effect. The plasmonic effects of metals can also serve as light encoders. When light interacts with a metal modulator, photons engage with plasmons, resulting in encoding. A practical demonstration of this concept is outlined in a study [91], wherein gold cells were deployed on a silicon substrate. Each cell housed smaller metal units with varied periods. These variations produced distinct response functions due to differentiated plasmonic effects, which in turn functioned as modulators.

### 3.8. Inverse Design of Encoder

Inverse design has emerged as a prominent approach in nanophotonics [92]. While its applications span various areas, the method has demonstrated significant potential for enhancing device performance. Notably, this strategy has potential applications in designing encoders for MCS. As previously highlighted, random structures can be employed as encoders; however, the efficiency and resolution may vary considerably across different random structures. Consequently, a designed structure may yield better outcomes and stability. However, because of the high complexity of the various degrees of freedom for design, direct system design becomes challenging. In this context, inverse design offers a promising solution to derive these structures.

Liu et al. [48] conducted an optimization of a random structure, building on the observation that certain “open channels” exist for each structure. This means that light transmission within nanostructures correlates with their wavefront. Leveraging this insight, they designed and optimized transmission efficiency for nanostructures. Similarly, Hadibrata et al. [93] adopted an optimization approach on a random structure, resulting in improved resolution and a reduced device size.

In recent developments, machine learning has been integrated into inverse design strategies [94]. Given that the design of nanophotonic encoders may encompass a vast array of parameters, machine learning is particularly suited to address this challenge due to its robust data processing capabilities. This fusion of machine learning heralds a new frontier in design methodologies. Specifically, encoder designs for MCS driven by machine learning have been explored.

In a study [95], researchers employed a generative adversarial network (GAN) to train and formulate the structure of nanostructures. The initial datasets were composed of simulated and fabricated structures with their respective transmission functions. The GAN was utilized to generate and validate individual new structures, from which the optimal one was selected for fabrication and use as an encoder. This workflow is depicted in Figure 6.

Similarly, Zhang et al. [12] adopted a comparable approach in designing encoders, with their focus centered on metasurfaces and thin films [96]. The nanophotonic encoder was conceived using parameters derived from the algorithmic encoder while retaining the decoder. Using these encoder parameters, various metasurfaces were designed. As a result of this methodology, they achieved superior accuracy and a thirty-fold increase in manufacturing tolerance, which is another important factor to note in the design.

### 3.9. Tunable Devices

In addition to the spatial multiplexing devices previously discussed, which often incorporate multiple detectors, temporal multiplexing offers another viable approach. Temporal multiplexing refers to devices that can be set to different configurations and then used to encode information for spectrometers. One of the most representative setups is the FTS, which can be implemented using different tuning methods. Conventional tuning mechanisms primarily encompass microelectromechanical systems (MEMS) and thermal tuning. However, recent advancements have introduced alternative tuning methods, such as liquid crystals and some innovative materials. According to the model presented in Section 2, as long as the device is tunable, we can form a transmission matrix A and perform computational recovery. FTS is an example using a Fourier matrix as the transmission matrix. But a number of modern, tunable MCS designs tend to conform to the principles of FTS. So, we first discuss the theory of FTS before discussing the tuning mechanisms used in MCS.

#### 3.9.1. Fourier Transform Spectrometer

The FTS is foundational in the history of computational spectrometry. Although the concept of MCS had yet to be formulated during its advent, the principles of FTS conform aptly to the modern criteria of MCS. In its nascent stages, FTS was conceptualized as an expansive system leveraging the MZI to manipulate phase differences, thus producing interference patterns. This process employs the tuning of OP, as illustrated in Figure 7. Intrinsically, the transmission matrix of an FTS aligns with a Fourier matrix, and the act of matrix multiplication simulates the execution of a Fourier transform. Consequently, an inverse Fourier transform facilitates the derivation of the spectrum.

The drive toward the miniaturization of FTS has been a significant research endeavor [97,98,99,100,101,102,103]. The primary strategy revolves around modulating the phase difference between light beams. An intuitive mechanism involves utilizing an MZI array where each individual MZI imparts a phase shift, serving as a discrete measurement point [103]. Such MZI array methodologies have later been instantiated through architectures like coiled spiral waveguides [104], advancements in sub-wavelength engineering [105], and the utilization of speckle dynamics [106]. However, these techniques inherently follow a spatial multiplexing paradigm, resulting in an extensive footprint. As alluded to previously, a more preferable way of implementing FTS is through the integration of tunable mechanisms, as evidenced by propositions of thermally modulated FTS designs [107]. Examples are shown for each of the tuning methods mentioned below.

A distinct advantage of FTS is its augmented SNR, a merit attributed to its elevated throughput, colloquially termed Fellgett’s advantage [108]. Nevertheless, FTS presents a one-to-one correlation between measurements and spectral points, negating the advantages of compression. Parallelly, akin to the earlier-addressed waveguide-based MCS, the resolution determinant in an FTS is the maximal OP differential between the bifurcated light beams, a parameter that might pose challenges in extensive scaling.

FTS continues to dominate the discourse on the design of miniaturized spectrometers. The foundational principles of FTS have been harmonized with contemporary learning-based algorithms [109] and diverse material classes [99]. A number of new, inspiring designs have been proposed [110]. The academic community has offered a plethora of design methodologies, with exhaustive reviews such as [111,112] providing holistic perspectives. Given the extensiveness of these sources, this discourse abstains from further elaboration on the subject.

#### 3.9.2. Microelectromechanical Systems

MEMS technology stands as a pivotal method to mechanically tune devices, leading to discernible changes in their properties with a vast degree of freedom [113]. Renowned for its ability to modify device attributes, MEMS has been extensively implemented in applications like phase shifters [114], PhCs [115], and phase arrays [116].

Its potential has been explored in MCS as well [117,118,119,120,121]. Fathy et al. [122] utilized a MEMS actuator to actuate a micromirror, realizing an FTS scheme at the chip scale (Figure 8a). Similarly, Omran et al. [123] introduced an MZI strategy for MCS, leveraging an electrostatically actuated MEMS device. Beyond electrostatic actuation, MEMS devices can also be powered via electromagnetic [124] and electrothermal methods [125]. Non-FTS encoding can also be implemented. Qiao et al. [126] employed a MEMS device to adjust the vertical separation (Figure 8b), thereby altering the coupling between two strip waveguides. By varying the voltage applied to the MEMS device, different distances and subsequently different couplings are achieved. Each specific position can function as an encoder, with the encoding matrix derived through calibration using a tunable laser source.

While MEMS devices are good at handling many measurements in a small space, they face issues with consistency [127]. Positional accuracy may not be consistent, which can alter the transmission matrix, thereby introducing spectral inaccuracies.

#### 3.9.3. Thermal Tuning

Thermal tuning is also an important way to adjust devices. Specifically, materials like silicon show evident changes in how they modulate light when their temperature changes. By modifying the temperature, the desired light modulation can be achieved.

An FTS can be directly implemented by tuning the heat. In [6], an FTS is implemented using two arms of waveguides. The characteristics of both arms can be tuned thermally, and thus different Ops are attained. The detector records the interferogram, and the spectrum is recovered via inverse Fourier transform. The process is shown in Figure 9a. Thermal tuning remains a major tuning method in FTS [128].

Other than direct FTS implementation, a commonly used device is the resonator [107], for which the transmission peak changes with heat. In [129], a ring resonator is combined with a subsequent MZI structure. Two heaters are embedded within the ring resonator and MZI, which enable two tuning degrees of freedom. With the help of the two heat tuners, the spectrum can be sampled with different transmission functions. Similarly, Sun et al. [108] showcased an on-chip spectrometer built on a double-side-coupled grating-assisted Fabry-Pérot array [80] utilized as unit cells. They designed arrays with both 5- and 7-unit cells, each operating within distinct spectral bands. Subsequent thermal tuning was applied to each unit, aiming to adjust the resonant wavelength. Through a combination of thermal tuning and cascading techniques, the device achieved impressive resolutions of better than 0.43 nm and 0.51 nm, covering spectral bandwidths of 73.2 nm and 102.7 nm for the 5- and 7-unit configurations, respectively. Visualizations of both designs are shown in Figure 9b,c.

A large-scale tuning network structure has also been studied. Yao et al. [130] used a mesh of thermally tuning MZIs and pushed the resolution to 20 pm with a bandwidth of 115 nm. Also, Xu et al. [131] used an array of thermally tunable cavities formed by waveguides. Each channel generates a distinct speckle, and with the help of tuning, the usable channel number is high, which pushed the resolution to 5 pm with more than 2 × 10^4^ channels.

#### 3.9.4. Other Tuning Methods

Besides MEMS and thermal tuning, many other tuning mechanisms can also be studied and applied [132,133,134]. Ni et al. [135] introduced a device employing tunable liquid crystals to encode light, facilitating the concurrent retrieval of both the spectrum and polarization state. As shown in Figure 10a, their design incorporates a 1D grating structure, infused with liquid crystal, and capped with a transparent indium–tin–oxide (ITO) electrode. With varying voltages, the liquid crystal undergoes rotation, resulting in shifts in the response curve. Aimed at supporting multiple high-Q resonances, the sensitivity of the device to both polarization and spectrum permits the extraction of both sets of information following an optimization process.

Meanwhile, Yuan et al. [136] suggested employing black phosphorus (BP) as a light encoder and integrating a spectral learning process to reconstruct the spectrum. The pronounced photoresponse and Stark effect [137] of BP are optimized in a device amalgamated with a graphene top gate [138] and trap-free hexagonal boron nitride (hBN). Remarkably diminutive in scale—approximately in wavelength dimensions—the design of the device takes the diffraction of light into account, potentially situating it among the most compact spectral sensing instruments. By modulating the voltage, the inherent biasing displacement field of the device undergoes changes, subsequently altering its spectral response. This response function of the device is obtained by utilizing a tunable blackbody radiation source, eliminating the need for optical components like lenses and gratings. Impressively, the device operates within a 2–9 μm wavelength range, maintaining a compact size of just 9 × 16 µm^2^. Such minuscule devices, when arrayed, pave the way for hyperspectral imaging.
Figure 10Other tuning methods for MCS. (**a**) A unit of tunable liquid crystal (figure reprinted with permission from [135] © 2022 Springer); (**b**) pipeline for single-dot perovskite MCS (figure reprinted with permission from [139] © 2022 Wiley).
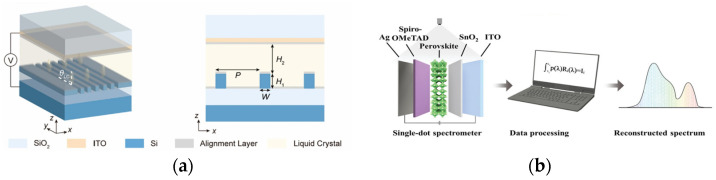



Yoon et al. [140] employed a tunable van der Waals (vdW) junction in the construction of a spectrometer. The hallmark feature of these electrically tunable vdW junctions is their controllable interlayer transport [141]. Such electrically modifiable transport facilitates the alteration of spectral response, showcasing remarkable sensitivity across an expansive spectral domain. A MoS_2_/WSe_2_ heterojunction, efficient across the visible to infrared spectrum, served as their junction of choice [142]. After the calibration of the spectral response curve, the captured spectrum is reconstructed through the application of convex optimization and regularization techniques. The heightened sensitivity of the spectral response curve allowed for impressive specifications, achieving a resolution nearing ~3 nm within a spectral range of ~405 to 845 nm.

Moreover, perovskite has also been explored as a medium for spectrometers. The hybrid organic-inorganic perovskite is renowned for its elevated carrier mobility and its tunability [143]. Guo et al. [139] implemented a perovskite layer, with its state controlled via a bias voltage, to engender multiple spectral responses. Their innovative in situ modulation approach permitted a significant reduction in device size. Its illustration can be found in Figure 10b. In a parallel development, Sun et al. [90] harnessed electron donor controls to achieve a more rapid response coupled with enhanced resolution. Meanwhile, Zhang et al. [144] put forward the idea of using a bandgap-gradient perovskite in conjunction with spectrum projection for color sensing applications.

The various types of encoders have been discussed extensively in the previous sections, with their advantages and disadvantages highlighted. Table 1 briefly compares the different encoders used in MCS.

## 4. Decoder

In computational spectrometers, the number of measurements is less than the number of data points recovered. Thus, it is an ill-posed problem, and the transmission matrix has more columns than rows. To retrieve the original spectrum, multiple computational recovery methods have been experimented with.

### 4.1. Singular Value Decomposition

SVD is used to factorize a non-square matrix to obtain the singular values. A number of previous research studies have used this method to recover spectrums from measurements. A matrix of size *mn* can be decomposed as follows:(5)$$M_{m\times n}=U_{m\times m}\sum_{m\times n}V_{n\times n}^{*}$$
where $U_{m\times m}$ is the left square matrix, and $V_{n\times n}$ is the right square matrix. The * stands for the Hermitian transpose of $V_{n\times n}$. As all the values in the spectrum are positive and real, we can use the transpose of $V_{n\times n}$.

The inversion of the original matrix $M_{m\times n}$ can be derived using the inversion of each of the decomposed matrices. As both $U_{m\times m}$ and $V_{n\times n}^{*}$ are square orthogonal matrices, a direct inversion can be carried out. $\sum_{m\times m}$ is a diagonal matrix containing only the singular values and with zeros as the non-diagonal values. Thus, the inversion of $\sum_{m\times m}$ can be constructed as the inversion of each of the values on the diagonal. The inversion of $M_{m\times n}$ can be written as follows:(6)$$M_{m\times n}^{†}=V_{n\times n}\sum_{n\times m}^{†}U_{m\times m}^{T}$$
where $M_{m\times n}^{†}$ stands for the pseudo-inversion of $M_{m\times n}$, and $\sum_{n\times m}^{†}$ is essentially the element-wise inverse of $\sum_{m\times n}$. Thus, we have an approximation of the inversion of the matrix, which we can pseudo-inverse. Thus, the spectrum can be obtained via inversion. This method has proved successful in various setups.

The singular values are ordered by magnitude in matrix $\sum_{m\times n}$, and while larger singular values stand for major modes in the original signal, smaller ones usually contain a major amount of noise. By only selecting the large singular values and writing off the smaller ones, the noise can be suppressed to enhance the accuracy of the reconstruction. This “truncation” method is widely used in many cases and allows for improved accuracy and speed [145,146,147]. The optimal truncation has also been analyzed [147,148], guiding the selection of singular values.

### 4.2. Convex Optimization

While SVD works well in many systems [96], convex optimization is more commonly used in CS problems [149]. A convex optimization problem is formed as follows:(7)$$X=\underset{x}{\min}{\left|\left|Y-Ax\right|\right|}^{2}$$
where X is the estimated spectrum, Y is the measurement, and A is the transmission matrix. Typically, A is obtained from calibration. By minimizing the gap between the actual measurement and the supposed measurement, the spectrum can be obtained. As mentioned before, CS theory has proven that the signal can be mostly retrieved using a sampling ratio lower than the Nyquist sampling rate with convex optimization. Thus, convex optimization is now the most commonly used method. Many mature algorithms can be applied to find the optimal value of these convex problems, including the Newton conjugate gradient, the Broyden–Fletcher–Goldfarb–Shanno algorithm, sequential least squares programming algorithms, etc. Different algorithms vary in speed and accuracy, and they should be chosen based on the different problem settings. The implementation of these algorithms is also easy with the help of available optimization toolboxes such as SciPy [150].

### 4.3. Regularization

Convex optimizations consider the problem purely mathematical, and the spectrum recovery problem has its own internal structure, including the fact that the spectrum is often smooth and sparse. Thus, coupling prior knowledge can often help the optimization converge to a better minimum and/or make the process faster. Typical regularizations include the L1 norm, the L2 norm, and some specifically designed priors.

Adding the L1 norm, or Lasso norm, encourages sparsity as it pushes all the values of the target values to zeros. An L1 norm optimization problem can be mathematically written as follows:(8)$$X=\underset{x}{\min}{\left||Y-Ax|\right|}^{2}+𝜕||x||$$

This introduces a new hyperparameter of 𝜕 to control the importance of the sparsity, and tuning the parameter often results in a change in the minimum. The parameter can be empirically tuned based on different systems.

The sparse nature of spectrums has already been proposed and studied. In [151], Oliver et al. showed that the resolution can be improved hugely beyond the limitation of the number of filters. Through experiments, they have shown that the L1 norm is proper prior to being applied in the reconstruction of the spectrum. The study is moving forward to a deeper understanding of the nature of spectrums, and CS theory boosts the development of filter-based spectrometers.

On the other hand, adding the L2 norm, or namely, the Tikhonov norm, encourages the smoothness of the target value as follows:(9)$$X=\underset{x}{\min}{\left|\left|Y-Ax\right|\right|}^{2}+\beta {||x||}^{2}$$

Similar to the L1 norm, the L2 norm also introduces a hyperparameter β that can be tuned based on different evidence from the system. The L2 norm naturally encourages the recovered result to be smooth, which caters to the fact that most spectrums are often smooth.

The advantages of both norms can be combined by using the weighted sum of both, termed elastic net regularization, as it behaves like an elastic net. This can be written as follows:(10)$$X=\underset{x}{\min}{\left|\left|Y-Ax\right|\right|}^{2}+𝜕||x||+\beta {||x||}^{2}$$

The elastic net methodology, although designed to combine the advantages of both L1 and L2 norms, introduces two additional hyperparameters. This expansion in the hyperparameter space not only amplifies the complexity of parameter tuning but can also induce instability in locating the optimum value. Nevertheless, both L1 and L2 regularizations are extensively employed in various contexts and have led to significant enhancements in numerous scenarios.

Beyond the widely acknowledged properties of spectrums being both sparse and smooth, certain intricate characteristics are also employed to instill a more nuanced prior. The work in [56] leveraged a Gaussian basis, predicated on the premise that the image point embodies a Gaussian configuration. Initially, these points are transformed into the Gaussian basis. The ensuing optimization is executed within this Gaussian basis before the final results are reverted to the original space.

It is vital to note that hyperparameters typically exhibit a dependency unique to each system. Modulating these parameters can profoundly influence the eventual outcome. Techniques such as the L-curve and generalized cross validation (GCV) offer avenues for hyperparameter tuning. While the L-curve is primarily geared toward enhancing the performance of the current dataset, GCV is oriented toward optimizing unseen data samples. Additionally, it is pertinent to highlight that SVD is mathematically congruent to convex optimization with regularization.

### 4.4. Dictionary Learning

The sparsity of the signal is a widely used regularization and achieves a large improvement in the result. But some of the spectrums are not naturally sparse, which violates the prior. To find a basis that can mostly sparsely represent the original signal, multiple bases are proposed, such as the Gaussian [152], Lorentzian, and secant hyperbolic basis [151]. However, these designed kernels have limited ability to find the optimal sparse representation. Dictionary learning has been applied in image and signal processing since it was proposed [153,154,155,156]. The basic idea of dictionary learning is to learn the matrix composed of the major modes of the collected data, which is to find a certain set of bases that best sparsely represents the distribution of data [157].

Zhang et al. [158] proposed to use dictionary learning in the recovery of MCS. They formulated the problem as a minimization in a new dimension:(11)$$\underset{x}{\min}\;{\left\|x\right\|}_{1}$$
$$Subject\;to\;{\left\|AØx-y\right\|}_{2}<\varepsilon$$
where x is the spectrum to be recovered, y is the measurements, A is the transmission matrix, and Ø is the sparse basis transformation. Ø is derived by collecting a dataset and then learning from the dataset using an optimization scheme. The optimization problem is solved in two steps, as there are two unknown variables:(12)$$\underset{T,Ø\;}{\min}{\left\|T-ØD\right\|}_{F}^{2}$$
$$Subject\;to\;\|d_{i}\|{}_{0}\leq \tau \;\left(i=1,2,3,\ldots ,p\right)$$
where T is the training set, and D is the sparse representation matrix. τ is the constraint on sparsity for each element $d_{i}$ in matrix D. The Frobenius norm is defined as ${\left\|A\right\|}_{F}^{2}=trance\left(A^{T}A\right)$. As this optimization has two target variables, the optimization is carried out in two steps:
1.Sparse approximation: the dictionary Ø is kept and optimizes the sparse representation D;2.Dictionary update: the dictionary Ø is updated after getting the sparse representation.

They showed a result that surpasses the Gaussian basis, and it coincides well with the ground truth. Besides the dictionary learning method mentioned, a few variants of dictionary learning are also proposed, such as task-driven [159], Bayesian-based [160], and kernel-based learning [161].

### 4.5. Deep Learning

In many MCS, the transmission matrix is typically derived through calibration. The prevailing methodology leans toward CS, wherein the number of measurements is fewer than the recoverable data points. A notable challenge is the potential inaccuracy of the calibration matrix in representing the system configuration, primarily due to noise perturbations.

Deep learning (DL), as shown in [162], has garnered immense attention in arenas such as image processing [163,164] and pattern recognition [165,166]. Contrary to linear algorithms, DL is adept at modeling intricate non-linear systems. Through meticulous training, it discerns an optimal basis within this non-linear domain, thereby facilitating efficient signal representation and enhancement in resolution. For instance, Kim et al. [167] utilized the UNet model [168] for signal recovery in multi-thin-film measurements, effectively capturing multiple peaks and broadband spectrums. Chatzidakis et al. [169] innovated DL methodologies to rectify calibration drifts in MCS systems over long durations. Such incorporation of DL into inverse design is fundamental, signaling a shift toward a decoder-centric paradigm [95,170]. Furthermore, its prowess extends to spectral analysis, as exemplified by Said et al. [171], who employed DL to analyze spectra from milk samples, gleaning vital information about its constituents.

The superiority of DL is particularly pronounced in hyperspectral imaging, which is tasked with reconstructing compressed spatial and spectral data. Given the large dimensions of hyperspectral datacubes, conventional approaches like convex optimization can be either inefficient or computationally burdensome. DL, in contrast, adeptly models such domains and imbibes potent priors during training [172] to bolster resolution. Yet, its adoption for spectrum recovery remains nascent, with a predominant reliance on optimization methods.

Nonetheless, the inherent merits of convex optimization techniques, particularly their independence from extensive datasets, remain irrefutable. As highlighted in [173], evolving MCS systems over time may render the collection of vast datasets neither practical nor cost-effective. Conversely, optimization methods exhibit higher adaptability to system changes. An emergent trend involves amalgamating the data-neutral characteristics of optimization with the robust priors extracted from datasets. Leading strategies in this context encompass plug-and-play (PnP) [174,175,176] and untrained neural networks [151,152]. Conventionally, PnP techniques harness priors from datasets and weave them into optimization algorithms, similar to attributes like smoothness or sparsity. Such data-sourced priors have shown efficacy in varied applications, including denoising [177,178] and hyperspectral imaging [179]. Meanwhile, untrained neural networks, devoid of prior learning, are instrumental in matrix inversion. Leveraging their intrinsic capability in feature extraction, they frequently surpass many direct optimization methodologies.

## 5. Applications

While traditional bulky spectrometers have been developed as products and have found important applications in biosensing and consumer devices, MCS, because of its low cost, high resolution, and small footprint, will not only enhance traditional applications but also bring about new applications. Some of the major applications that MCS can be applied to are highlighted.

### 5.1. Industrial Applications

#### 5.1.1. Biosensing

Biosensing primarily refers to the sensing of molecules. As the absorption peaks of different types of molecules differ, the spectral footprint of molecules is one of the fundamental features for identification and analysis. The application of biosensing can be found in atoms, molecules, and at larger scales. MCS can serve as an important alternative in many areas because of its low cost and high resolution.

Bryan et al. [180] developed a silicon nitride microring resonator-based chip-wise spectrometer for biosensing. They tested it with a sucrose solution for bulk sensing and C-reactive protein. Guo et al. [181] developed an MCS based on plasmon PhC and tested it with a bovine serum albumin (BSA) protein layer and observed a resonance spectra shift caused by the BSA. More promising results [180,182,183,184,185] show the feasibility of biosensing with MCS. Also, MCS can be used to help with the design of other biosensors [186].

Spectrometers are widely used in medical analysis. While traditional tests with spectrometers are bulky and expensive, low-cost in situ diagnostics can be possible with the help of MCS. MCS have been demonstrated to help diagnose skin cancer [187] and conduct general health tests [187,188,189]. With the help of wearable MCS, personalized and preventive medicine can be developed [190].

#### 5.1.2. Chemical Sensing

Besides biosensing, MCS are vital in sensing chemicals, which turns out to be effective in multiple areas, including food safety, agriculture, and gas sensing. Optical spectrums have long been serving as an important indicator of the safety status of food. MCS enable quality checks to be much more accessible and affordable. Labaj et al. [191] developed a miniaturized mid-IR spectrometer to detect environmental changes. Additionally, MCS have found wide applications in various categories of food, including milk [192], meat [193], fish [194], etc.

Just like traditional spectrometers, MCS can be applied to agriculture. Bulky spectrometers have previously found a number of applications, including crop growth monitoring [195], plant vitality analysis [196], chlorophyll monitoring [197], and disease classification [198]. The use of MCS in agriculture has been proven recently. Shen et al. [199] used MCS to monitor the health status of soil. Kosmowski et al. [200] used MCS to identify cultivars and analyzed the data using machine learning. Franceschelli et al. [201] demonstrated the measurement of soil moisture. These experiments show the feasibility of applying MCS in agriculture and also indicate future development.

Gas sensing is another area where MCS can be applied. Gas analysis with a spectrometer has been demonstrated previously in a number of experiments [202]. Extensive research has been conducted on MCS-based gas sensing devices. In [203], Mannila et al. developed a cell-phone-based MCS sensor that can be used to detect carbon dioxide (CO_2_). Also, Erfan et al. [204] developed a MEMS-based MCS specifically for gas sensing in the 1300–1500 nm wavelength range, capable of detecting acetylene (C_2_H_2_), water vapor (H_2_O), and CO_2_. Muhiyudin et al. [205] developed MCS multi-gas sensing device with low power consumption. The sensor can be used to sense CO_2_, carbon monoxide, nitrous oxide, sulphur dioxide, ammonia, and methane. Similarly, gas sensing frameworks [206,207] have been proposed with enhanced performance.

### 5.2. Consumer Devices

MCS, due to their miniature size, low cost, and high resolution, can be another sensor equipped into consumer devices such as smartphones, watches, and other wearable devices. Spectroscopy on a cell phone has been previously demonstrated in the visible [208], near-infrared [209], and Raman [210]. Equipping smartphones with MCS allows users to capture spectral data easily, which can breed a number of new applications. Wearable devices can also benefit from MCS, as this adds a new degree of data collection, and the concept has been shown by researchers [173,189,211].

The aforementioned applications predominantly entail the substitution of conventional spectrometer techniques with considerably more affordable and attainable MCS. Numerous additional applications exist, although unaddressed within the scope of this discussion. The prospective market size for MCS is projected at approximately 900 million dollars [210]. As applications expand across various domains, it is anticipated that more and more will benefit from MCS.

## 6. Conclusions and Outlook

In this paper, we have surveyed the evolution of MCS, which leverages advancements in nanophotonics technology and computational methods. In contrast to traditional, bulkier spectrometers, MCS represent a significant leap in miniaturization while retaining high performance. The majority of current computational methods operate within the framework of CS theory, employing convex optimization techniques for the reconstruction of the original spectral signal. Consequently, our review primarily focuses on the design of nanophotonic devices, where most of the novel innovations are taking place. Our examination spans a range of technologies, from conventional waveguides to cutting-edge metasurfaces and innovative new materials. In conclusion, the advancement of MCS has garnered extensive research and has progressed swiftly. The performance of MCS has significantly improved, leading to numerous proficient applications. With further research, its performance and the breadth of applications will assuredly augment, contributing substantial value to the field.

Despite considerable progress in the field of MCS, several limitations remain unresolved. A primary issue concerns the trade-off between device footprint, bandwidth, and resolution. The physical size of the device limits the number of sensor units that can be incorporated, thus restricting the volume of data points that can be collected. The ensuing question is whether to allocate these data points over a wide frequency range to achieve broad bandwidth or to concentrate them within a narrow range for higher resolution. This essentially limits the information rate of the system.

In order to solve this problem, there are two immediate avenues for improvement. The first involves increasing the number of data points collected within a defined footprint, thereby enhancing the resolution and bandwidth of the device without having to physically enlarge the device. The second focuses on improving data representation efficiency so that the collected data are more meaningful and informative, thereby allowing for more accurate spectral reconstruction. Several potential methods are described below.

**Tunable devices**: Different from space-multiplexing, tunable devices multiplex in the time domain. This can largely improve data collection within the same footprint. As presented in the paper, MEMS, thermal tuning, and other tuning methods for multiple new materials have shown great potential for improved performance. There has been intensive research into exploring the usage of tunable mechanisms, and it remains a promising direction. However, the introduction of a tunable mechanism also brings about more concerns about the robustness of the system and may also increase the cost.

**Cascaded devices**: Cascading different devices can lead to a multiplication of different transfer functions, which may greatly improve the design freedom of the encoder in the system. Also, it may alleviate the problem of neighboring wavelengths having similar transmission functions and thus better differentiate wavelengths. Cascading different devices, especially tunable devices, may allow the system to function much better than their original devices. The combination of different devices with different characteristics, for example, FTS and ring resonators [95], may lead to a new level of performance.

**Advanced learning methods**: The algorithms used in MCS are dominated by CS theory, which is mostly based on the sparsity of the spectrum. But the convex optimization method used in CS may not be able to properly handle non-linearity in the problem. As mentioned previously, DL-based designs have recently emerged in spectral information acquisition, but most of them are used in hyperspectral imaging systems. One consideration might be the lack of a dataset for training, which remains a major issue. Another way is to generate new structures based on generative learning methods. Compared to random structures, which have already brought about many advantages, designing the structure of nanophotonic encoders using DL may better utilize the many degrees of freedom of the device, which contributes greatly to its performance.

In addition to enhancing performance, exploring new applications represents a promising direction in research. MCS systems not only exhibit performance comparable to traditional spectrometers, but they also have the advantage of being miniaturized and cost-effective. Beyond the conventional applications previously discussed, MCS have emerged in innovative application scenarios, including robotics and the Internet of Things (IoT). It is anticipated that the development and application of MCS will continue to flourish with increasing research efforts.

## Figures and Tables

**Figure 1 sensors-23-08768-f001:**
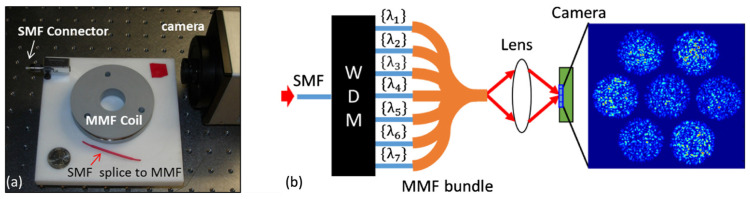
Fiber-based MCS. (**a**) MCS using a 100 m fiber (figure reprinted with permission from [27] © The Optical Society); (**b**) seven MMFs bundled in a spectrometer using a WDM for OCT (figure reprinted with permission from [28] © The Optical Society).

**Figure 2 sensors-23-08768-f002:**
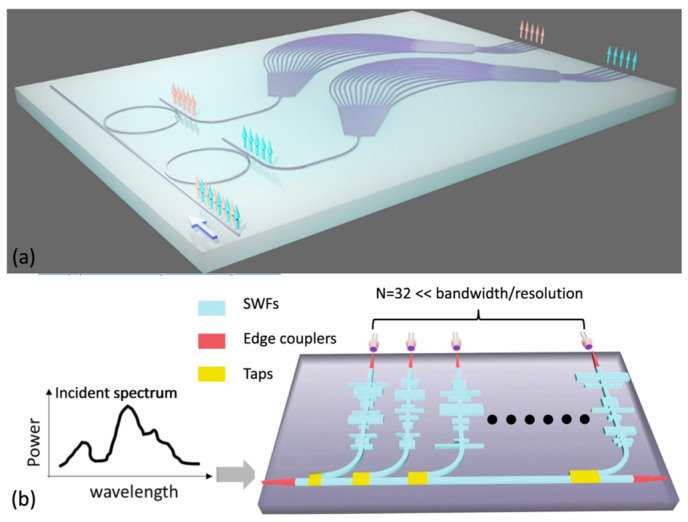
Waveguide-based MCS. (**a**) Tandem structure of ring resonator and arrayed waveguides (figure reprinted with permission from [38] © 2021 American Chemical Society). (**b**) Stratified waveguide as an encoder for the input spectrum (figure reprinted with permission from [40] © 2021 Nature Publishing Group).

**Figure 3 sensors-23-08768-f003:**
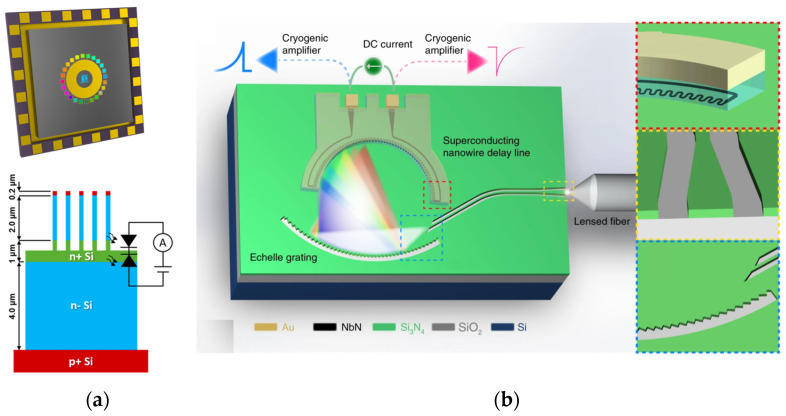
Nanowire-based MCS. (**a**) Nanowire combined with detector (top) and one unit of nanowire (bottom) (figure reprinted with permission from [57] © 2019 American Chemical Society); (**b**) superconducting nanowires as MCS (figure reprinted with permission from [58] © 2019 Nature Publishing Group).

**Figure 5 sensors-23-08768-f005:**
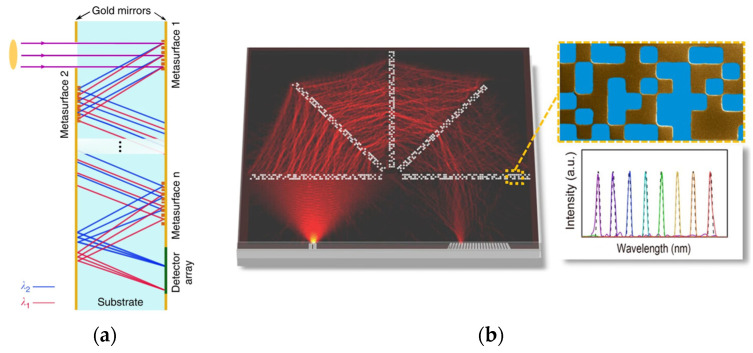
Metasurface-based MCS. (**a**) Folded metasurfaces for dispersive MCS (reprinted with permission from [87] © 2018 Nature Publishing Group); (**b**) folded digital metalens for MCS. Five metalenses are shown here, with exemplary designed structure and result spectrum (right) (reprinted with permission from [89] © 2023 American Chemical Society).

**Figure 6 sensors-23-08768-f006:**
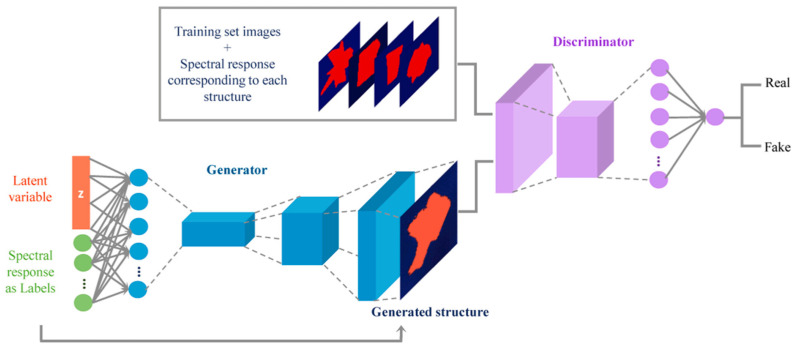
MCS based on inverse design (figure reprinted with permission from [95] © 2022 Elsevier).

**Figure 7 sensors-23-08768-f007:**
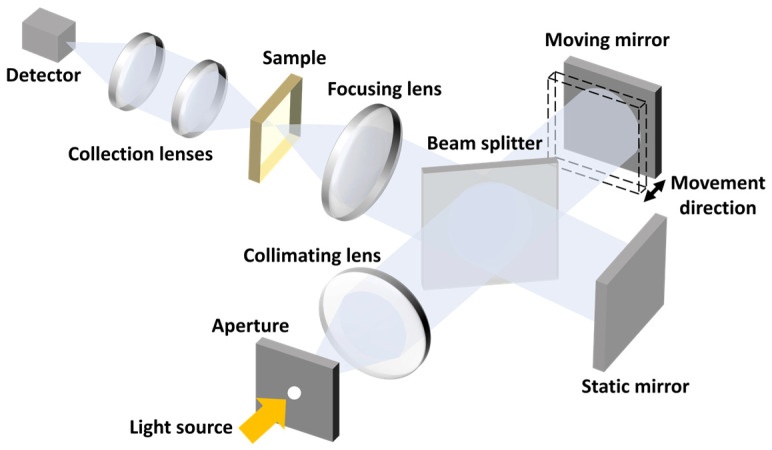
Principle of traditional FTS.

**Figure 8 sensors-23-08768-f008:**
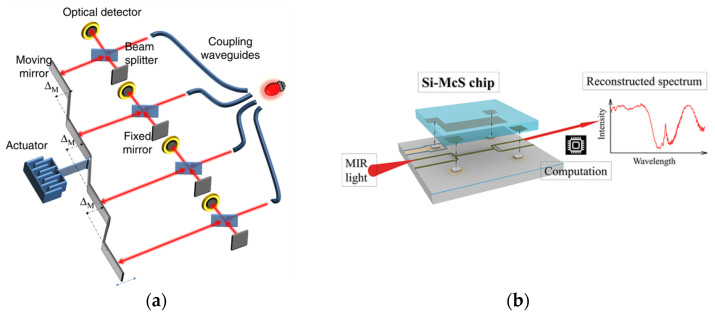
MEMS-based MCS. (**a**) MEMS used for an FTS. The actuator is used to move the mirror to create phase change (figure reprinted with permission from [122] © 2020 Nature Publishing Group); (**b**) MEMS device used to change directional coupling between waveguides (figure reprinted with permission from [126] © 2022 American Chemical Society).

**Figure 9 sensors-23-08768-f009:**
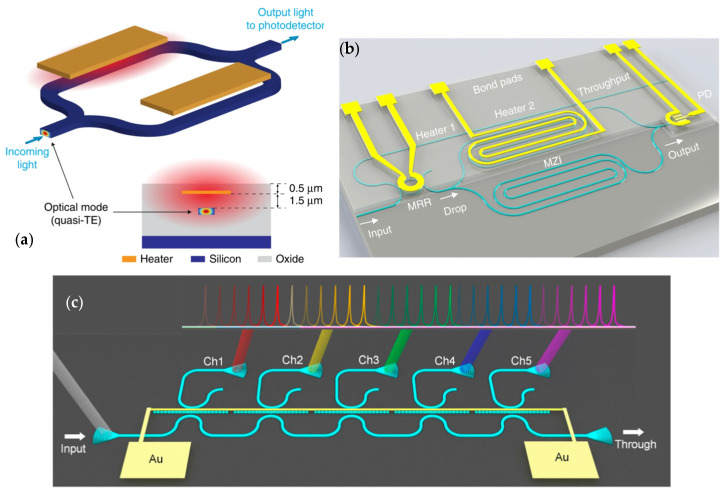
Thermal-tuning-based MCS. (**a**) An implementation of FTS using heaters to thermally create phase change for the FTS in [6] (figure reprinted with permission from [6] © 2018 Nature Publishing Group); (**b**) thermal-tuning-based ring resonator and FTS for MCS. Heater 1 changes the configuration of the microring resonator, and heater 2 changes the configuration of the waveguide to create phase change (figure reprinted with permission from [108] © 2022 American Chemical Society); and (**c**) integrated array of narrowband thermally tunable filters (figure reprinted with permission from [129] © 2019 Nature Publishing Group).

**Table 1 sensors-23-08768-t001:** Comparison of different encoders.

**Encoder**	**Typical Resolution**	**Typical Footprint**	**Advantages**	**Shortcomings**
Fiber	<1 pm [27]	cm scale	Low cost, high resolution	Low-level miniaturization
Waveguides	<10 pm [42]	35 × 260 µm^2^ [42]	Low cost, structure variability	Relatively larger footprint
Random structure	<0.75 µm [46]	25 µm radius [46]	High resolution, smaller footprint	Large loss for scattering, structure uncertainty
Nanowires	<10 nm [56]	0.5 × 75 µm^2^ [56]	Very small size, easy to combine with CCD	Relatively lower resolution
PhC	<0.1 nm [64]	0.6 × 114 µm^2^ [64]	High resolution, cost effectiveness	Manufacture error sensitivity, larger footprint
QD	<1 nm [75]	cm scale, compatible with CCD	Easy to combine with CCD, small footprint	Similar responses function for QDs restrict resolution
Meta-surfaces	<0.8 nm [85]	cm scale, compatible with CCD	Large design freedom, structure variability	Relatively lower resolution
Tunable devices	<5 pm [131]	16 × 9 µm^2^ [136]	High resolution, small footprint	Environment sensitivity, more sophisticated manufacturing

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
