# Peer review of "Review of Miniaturized Computational Spectrometers"

_sensors, 2023, doi:10.3390/s23218768_

Round 1
Reviewer 1 Report
Miniaturized computational spectroscopy (MCS) has received widespread attention due to its applications in medicine, biosensing, etc. In this review, the authors outlined the principle of MCS and compared different encoding strategies. Then, they explored convex optimization and machine learning decoding algorithms. At last, the authors provided examples of a wide range of MCS applications and discussed the outlook of MCS.
This review was finished with very high quality, which means it can almost be published in the current version. However, some detailed errors in this review still need to be carefully corrected before publication. If the following problems are well-addressed, this reviewer believes that the essential contribution of this review is vital for the research of Miniaturized computational spectroscopy.
- For every Figure that appears in the review, the author needs to provide a corresponding correspondence in the manuscript. For example, Figure 1b corresponds to line 171; Figure 2a corresponds to line 202; Figure 2b corresponds to line 222.
- In the final version, the authors need to make sure that each figure is centered, as in the review version they were displayed left.
- In line 353, is it suitable to use ‘-’ at both sides of ‘1D, 2D and 3D’?
- In Chapter 3, the author conducted a detailed classification of the literature on encoder. But as a review, it is best to make a summary at the end of the chapter to add your own thinking at a higher level.
- Authors need to unify the format of all references, such as some references lack DOI (Refs. 2, 4, 24, 25, 31, 32, 33 ), and some journal names do not have abbreviations (Ref. 97).
Reviewer 2 Report
This review paper provides a comprehensive and well-structured overview of the recent developments in MCS. The paper effectively summarizes the major MCS designs and computational methodologies, offering a clear comparison of different structural models. Additionally, the discussion on future research directions and potential applications adds depth to the review. Overall, this paper is a valuable resource for researchers and professionals interested in the evolving field of MCS, and I recommend accepting it for publication.
Reviewer 3 Report
1. Structure of the study to be added
2. The compartive results to be added
3. Conclusion is not clear
4. The limitations of the study to be added
5. The grammatical errors to be checked
6. Review and add the following reference, S. Nallusamy and K. Sujatha, (2021), “Experimental analysis of nanoparticles with cobalt oxide synthesized by coprecipitation method on electrochemical biosensor using FTIR and TEM”, Materials Today: Proceedings, Vol. 37, No. 2, pp. 728-732
To be checked
Reviewer 4 Report
The present review article, authored by Q. Guan, Z.H. Lim, H. Sun, J.X. Chew, and G. Zhou, adresses the development of miniaturized computational spectrometers (MCS), primarily focusing on the design of nanophotonic devices, and briefly bridging theory to actual application.
The review is written in clear English, and each section of the review is well organized. The illustrations also help the reader to understand the different architectures of nanophotonic devices. Computational spectrometers in general represent a hot and very up-to-date topic, and the combination of a theoretical approach and a variety of applications in one review paper seems very interesting to me and may stimulate new ideas for development. My only complaint of this review is the Section 5.1. titled “Biosensing”, as the authors give an overview of MCS in food industry, agriculture and gas sensing in addition to biomedical applications. It would be beneficial to rename the title of this Section, and/or further elaborate/cite more the specific application. If possible, please indicate which specific gas is described in references 201-204.
The conclusions seem sound and the references are appropriate.
